# SatGuard: Concealing Endless and Bursty Packet Losses in LEO Satellite Networks for Delay-Sensitive Web Applications

Submission Id: 1935

## ABSTRACT

Delay-sensitive Web services are crucial applications in emerging low-earth orbit (LEO) satellite networks (LSNs). However, our real-world measurement study based on SpaceX's Starlink, the most widely used commercial LSN today reveals that the endless and bursty packet losses over unstable LEO satellite links impose significant challenges on guaranteeing the quality of experience (QoE) of Web applications. We propose SatGuard, a distributed in-orbit loss recovery mechanism that can reduce user-perceived delay by completely concealing packet losses in the unstable and lossy LSN environment from endpoints. Specifically, SatGuard adopts a series of techniques to: (i) correctly migrate on-board packet buffer to support link local retransmission under LEO dynamics; (ii) efficiently detect packet losses on satellite links; and (iii) ensure packets ordering for endpoints. We implement a SatGuard prototype, and conduct extensive trace-driven evaluations guided by public constellation information and real-world measurements. Our experiments demonstrate that, in comparison with other state-of-the-art approaches, SatGuard can significantly improve Web-based QoE, by reducing: (i) up to 48.3% of page load time for Web browsing; and (ii) up to 57.4% end-to-end communication delay for WebRTC.

## 1 INTRODUCTION

We are experiencing two major changes in today's Internet. Upper layer applications are marching to the new era of Web 3.0, driven by a range of new technologies [34, 40]. Simultaneously, the underlying network architecture is evolving from the land to outer space, stimulated by the recent low earth orbit (LEO) satellite constellations. Big companies, such as SpaceX [16], Telesat [17] and Amazon [6], are planning and deploying satellite constellations (*e.g.,* Starlink and Kuiper) for global Internet access. These constellations, consisting of a large number of broadband satellites equipped with laser inter-satellites links (ISLs) [1, 14], ground-satellite links (GSLs), and user-satellite links (USLs), extend the boundary of today's Internet and construct LEO satellite networks (LSNs) to provide low-delay and high-speed Internet services ubiquitously [31, 37].

Web-based applications should be important scenarios in the upcoming era of satellite Internet. Representative Web applications, such as Web browsing and Web-based real-time communication (*e.g.,* WebRTC [18]) are expected to provide good end-user quality of experience (QoE) under LSNs. However, our real-world measurements on Web applications in the operational Starlink LSN plot a gloomy picture: while the low orbit altitude guarantees low baseline propagation delay for LSN users, the inherent unstable and lossy satellite links impose significant challenges on sustaining good Web QoE. Since LEO satellites continuously move at high velocity related to the earth surface, such inherent LEO dynamics can result in *endless and burst* packet losses for transport endpoints. Although the existing sender-side reliability control used in *statue quo* Web applications can recover lost packets, retransmission from

the sender inevitably involves extra delay and high jitter which can further hinder user-perceived Web experience (*e.g.,* prolonging Web page load time and user-perceived communication delay).

Existing approaches to accomplish fast loss recovery (and thus reduce the user-perceived delay) can be classified into two major categories. On one hand, many efforts [28, 32, 47, 61] proposed to use network coding (*e.g.,* forward error correction, FEC) to inject redundant information in data packets on the source, and recover the lost packets on the receiver without the need of sender-side retransmission. However, due to the highly dynamic packet loss ratio, a lower redundancy coding information may lead to recovery failure while a higher redundancy one requires much additional bandwidth, which is scarce in LSNs. Although recent studies have used machine learning technologies to predict packet loss rate [32, 41], the used datasets can not reflect unique packet loss behaviors in LSN scenarios (*e.g.,* handovers in satellite systems and signal attenuation caused by objects blocking the view of satellite terminals). On the other hand, other works proposed link-local retransmission (LL-ReTx) to buffer packets on intermediate network nodes, and re-send lost packet locally. This idea is mainly implemented in either (i) the transport layer [20, 22, 38, 42, 51] (*i.e.,* intermediate nodes have full transport-layer functions and maintain information for each flow, thus splitting the end-to-end connection into segments) or (ii) the link layer [10, 35, 52] (*i.e.,* only buffering packets or frames at the network interface and retransmitting them when any loss is detected). However, due to the high LEO dynamics, frequent disconnections and reconstructions of satellite links can disrupt the retransmission process and limit the recovery efficiency.

In this paper, we propose SatGuard, a novel in-orbit loss recovery mechanism that can be deployed in operational LSNs to effectively improve QoE for Web applications. SatGuard extends the existing link local retransmission mechanism to address the unique challenges caused by LEO dynamics and conceal packet losses for endpoints. The core design of SatGuard incorporates three key techniques: (i) leveraging predictable handover information (*e.g.,* ground-satellite connectivity and accurate handover time) based on the global scheduler of LSN operators to correctly migrate buffered content; (ii) dynamically configuring appropriate packet loss detection parameters (*i.e.,* timeout for LL-ReTx) for land-based users served by different satellite beams to improve recovery efficiency, and (iii) preserving the order of packets while leaving LSNs without head-of-line blocking, which can reduce the negative impacts of the disordered packets generated by LL-ReTx (*e.g.,* increasing jitter buffer greatly) on the upper-layer protocols.

We have implemented a SatGuard prototype upon Linux, and built an LSN simulation environment based on public constellation information and realistic Starlink data trace. SatGuard only requires operator-side modifications, and can be deployed in operational LSNs by software upgrades on satellites, ground stations

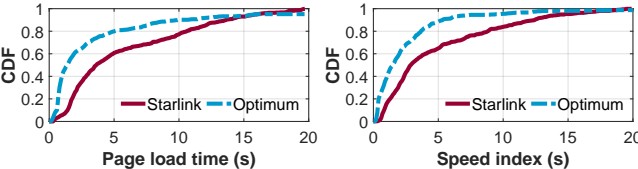

Figure 1: A high-level architecture of today's LSNs.

and satellite terminals, without the need to change user-side application interface. Our extensive evaluations demonstrate that: in comparison with other existing reliability control approaches, SATGUARD can: (i) reduce 99th percentile end-to-end delay by up to 59.8% and keep higher capacity utilization on average; and (ii) speed up Web browsing time by up to 48.3% and reduce user-perceived communication delay by up to 57.4% for WebRTC sessions.

In summary, main contributions in this paper can be concluded as follows: (i) we conduct a real-world measurement study to identify and analyze the performance issues suffered by representative Web applications in emerging LSNs; (ii) we present SATGUARD, an in-orbit fast loss recovery mechanism which extends existing LL-ReTx mechanisms to accomplish consistent low delay in unstable and lossy LSNs and improve user-perceived QoE; and (iii) we implement a SATGUARD prototype and conduct extensive, real-data-driven evaluations to verify the improvement on Web browsing and RTC. We will release the source code of SATGUARD.

## 2 BACKGROUND AND MOTIVATION

### 2.1 LEO Satellite Networks

Figure 1 plots a typical networking architecture that has been widely used by today's commercial LSNs such as SpaceX's Starlink [4]. At a high level, an LSN contains a *space segment* with a large number of LEO broadband satellites, and a *ground segment* consisting of geo-distributed ground stations (GSes) and satellite terminals (dishy) [8, 9, 13, 15]. Satellites can be equipped with laser inter-satellite links (ISLs) for inter-satellite communication, be equipped with radio ground-satellite links (GSLs) for ground communications, and further be connected to satellite terminals via radio user-satellite links (USLs) consisting of multiple spot-beams. When a user terminal accesses Internet services (*e.g.,* a Web server) through the LSN, the user's request is first forwarded to a ground station through one (*i.e.,* via bent-pipe transparent forwarding [46]) or multiple satellites (*i.e.,* via ISL-based space routing [44]) and then to a Point-of-Presence (PoP) of the Internet. Similarly, the server's response returns to the user terminal via the reverse path.

### 2.2 Understanding Web Performance in LSNs

To quantitatively understand the user-perceived experience of typical Web applications in emerging LSNs, we conduct a real-world measurement study in Starlink, the currently most popular commercial LSNs with more than 2 million subscribers globally [57].
**Experiment setup.** We deploy a vantage point with two laptops in New York City (NYC). These two laptops connect to the Internet via Starlink and Optimum (*i.e.,* a wired broadband service provider

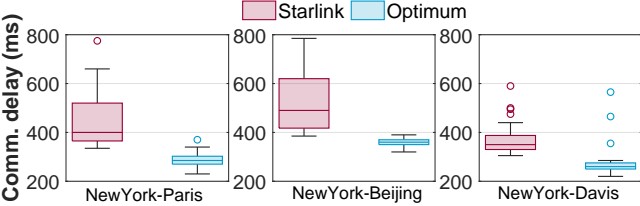

Figure 2: QoE analysis for Alexa Top 200 Web browsing.

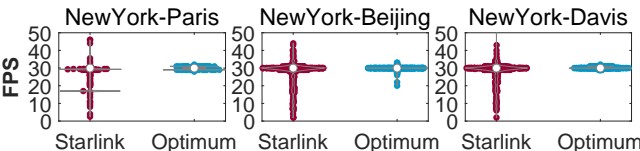

Figure 3: The end-to-end user-perceived communication delay for WebRTC communications.

Figure 4: Violin plot of FPS in WebRTC communications.

in U.S.) respectively. Further, we measure the QoE of two representative applications: Web browsing and Web Real-Time Communication (WebRTC) via the two different networks. For Web browsing, we use Chrome to visit Alexa Top 200 websites [5]. We then exploit the Browsertime tool [7] to measure: (i) *page load time*, which refers to the time it takes for a Web page to fully load in a browser, and (ii) *speed index*, which measures the visual completeness of a Web page, and lower values indicate faster perceived page loading. For WebRTC, we use Kurento [11] to build RTC sessions between the laptop client in NYC and other users in Paris (France), Beijing (China) and Davis (U.S.). Based on WebRTC's statistics APIs [58], we measure: (i) end-to-end user-perceived communication delay; and (ii) frames per second (FPS). In addition, we use tcpdump to collect packet-level traces of the two applications for further network analysis.

**Observations.** Figure 2 plots the QoE results of Web browsing in different networks. We observe that in the Starlink network environment, both page load time and speed index are much higher than those in the wired network. Specifically, the 60th/80th/90th percentile page load time in Starlink are about 245.0%/202.4%/136.3% higher than those in Optimum. Similarly, the 60th/80th/90th percentile speed index in Starlink is about 185.3%/241.6%/219.3% higher, indicating that for the same Web content, LSN users suffer from higher perceived delay and worse browsing experience. Since high page load time can increase the likelihood of users leaving the website [59], optimizing the page load time in LSNs should be an important goal for both network operators and content providers.

In addition, Figure 3 and Figure 4 show the QoE analysis for WebRTC in different networks. We find that for all three sessions associated with users in different locations, they experiences higher user-perceived communication delay and fluctuating FPS in Starlink, indicating a worse user experience compared to the wired

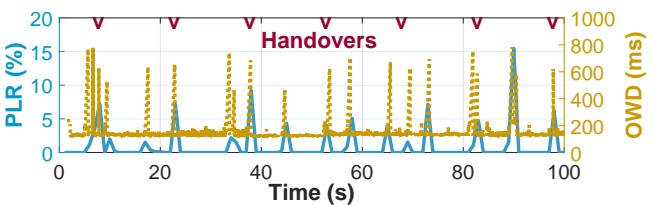

**Figure 5: Network-level burst packet loss rate (PLR) and one-way delay (OWD) variation measured in Starlink.**

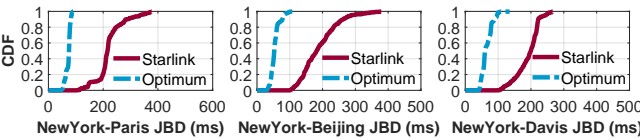

**Figure 6: JBD analysis for WebRTC communications.**

network. Specifically, as shown in Figure 3, the average communication delay experienced in Starlink is about 57.2%/43.1%/34.5% higher than that in Optimum for the three sessions. As shown in Figure 4, we observe that Optimum achieves a stable FPS with an average value of 30 and smooth user experience, while the FPS in Starlink is very relatively unstable and changes frequently.

**Root cause analysis.** To uncover the underlying factors that hinder Web QoE, we further conduct an in-depth analysis. Based on the traces collected by `tcpdump`, we calculate the network-level packet loss rate (PLR) and packet-level one-way delay (OWD), which refers to the duration from the first time the data packet is sent by the sender, to the first time it is received by the receiver. We reveal two LSN-specific factors that *jointly* affect the user-perceived Web QoE.

**(i) Endless and bursty packet loss in dynamic LSNs.** Figure 5 plots the PLR over time experienced by the Starlink client and a website server. We find that the bursty PLR ranges from 3% to about 15%. Every 3-5 seconds, a sudden increase in PLR can be clearly observed. Unlike traditional terrestrial Internet where the key network infrastructure is sealed in a protected environment, the core space backbone (*i.e.,* a large number of satellite switches or routers) of an LSN is operated and exposed in the *error-prone intermittent space environment*, in which packet loss may occur *frequently* for both ground-satellite and inter-satellite communication due to a ranges of factors such as LEO dynamics [21, 36] and electromagnetic interference [33, 45, 46]. For example, recent reports have demystified the Starlink's handover strategy [29, 55, 56], which uses a global scheduler to manage the satellite-terminal connectivity. Handovers occur every 15 seconds between the satellite terminal and its ingress satellite. In Figure 5, we also plot the time point when the handovers occur during the Web session. We observe that every time a handover occurs, it is always accompanied by a significant and noticeable increase in PLR.

**(ii) Long tail delay due to sender-based retransmission.** To cope with packet losses, existing approaches typically use *sender-based retransmission* to guarantee reliable transmission in lossy networks (*e.g.,*TCP retransmission). However, sender-based retransmission can result in *out-of-order packets and long-tail delay* in the lossy LSN environment. We also plot the OWD over time in Figure 5, and we find that burst high delays are always accompanied by burst high packet losses. Over the entire session, although the low orbit altitude enables low propagation delay over satellite links, we still observe that the 99th percentile receiver-perceived delay can reach up to 523.3% of the medium value. These high delay variations can seriously affect Web user experience. On one hand, a Web page typically consists of a number of objects (*e.g.,* images, texts and scripts), and the total page load time mainly depends on the last object received. On the other hand, Web-based RTC applications typically use a receiver-side jitter buffer to ensure that the audio

and video are played smoothly. The jitter buffer size is determined based on the maximum delay observed by the receiver over a past period of time. Therefore, the high tail delay can increase the jitter buffer delay (JBD) on the receiver, and accordingly amplify the user-perceived end-to-end communication delay. Figure 6 plots the JBD results of the three WebRTC sessions in previous experiments. We find that the burst packet losses and high delay variation indeed increase the JBD for RTC sessions in Starlink, which finally increase the end-to-end communication delay.

## 2.3 Why Existing Approaches Are Insufficient?

The network community has a long history of research into fast packet recovery, which can be mainly divided into two categories. **Network coding.** One classic technique for fast packet recovery is Forward Errors Correction (FEC) [28, 47, 61]. FEC works by adding redundant information to the original data packets before transmission. These redundancy packets contain additional information that allows the receiver to recover lost or corrupted packets. When a packet is lost or damaged in transit, the receiver uses the redundancy packets to reconstruct the missing or corrupted data, thus achieving fast packet recovery without sender-side retransmissions.

However, as shown in Figure 5, packet loss rates in LSNs are highly dynamic. FEC typically applies a fixed level of redundancy to all packets, or simply make overdue redundancy adjustments based on recently perceived network conditions. Selecting the right level of redundancy can be quite challenging in LSNs. Adding more redundancy to the original data packets improves error recovery capabilities, but significantly increases bandwidth overhead which is a scarce resource in LSNs. Although machine learning technologies can help to train a model that can predict packet loss rates by inputting a large number of real-word network measurement data [32, 41], the effectiveness is limited by the quality of the dataset and end-device computing capacity to run a heavy neural network model quickly and get accurate results. Besides, to our knowledge, there is no large-scale dataset currently that can reflect the unique packet loss behaviors of emerging LSNs described in §2.2.

**Link local retransmission (LL-ReTx).** Another existing method for fast packet recovery is to retransmit lost packets directly at intermediate nodes instead of requiring the sender to retransmit. Many previous efforts deploy buffers on intermediate node to realize such link local retransmission [35, 52], which can greatly shorten the recovery delay of lost packets and save a lot of bandwidth. However, these solutions are originally designed for static networks such as data center networks where intermediate nodes with stateful buffers are relatively static. The relative positions of intermediate nodes in the LSNs change rapidly over time, making it challenging to directly apply existing LL-ReTx mechanisms in LSNs, because the upstream node which buffers the lost packets may not connect to the downstream node after link handovers. Although some methods attempt to deploy full transport-layer functions in intermediate nodes to achieve a hop-by-hop transmission [19, 38, 42],

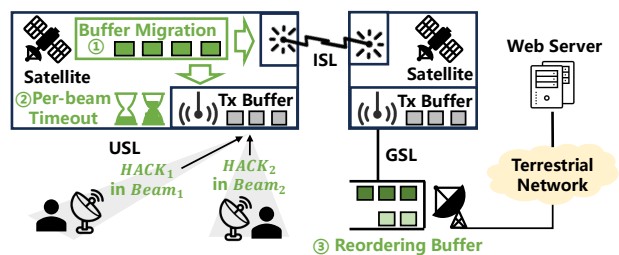

**Figure 7:** SatGuard **design overview, where green components are key techniques proposed in** SatGuard**.**

which can also recover packets loss in dynamic scenarios after new transport-connection built. However, heavy functions and large per-flow transmission states pose huge challenges for satellites with scarce resources. Moreover, frequent connection reconstruction may not meet the requirements of persistent delay-sensitive Web application (*e.g.,* WebRTC).

**Takeaways.** Collectively, our measurement study and analysis demonstrate that due to the unique LEO dynamics and lossy links in LSNs, end-to-end communications suffer from consistent, frequent, and highly-dynamic packet losses, which further affects the user experience of upper-layer delay-sensitive applications, such as Web browsing and WebRTC. Existing fast recovery methods either could cause extra waste of scarce bandwidth resources in LSNs, or are not applicable for LSNs with unique dynamic topology. The *statue quo* motivates us to seek an effective solution to cope with such LSN loss and improve Web user experience.

## 3 SATGUARD **DESIGN AND IMPLEMENTATION**

We present SatGuard, a novel in-orbit loss recovery mechanism that can effectively improve user-perceived Web QoE by enabling fast link local retransmission and concealing endless and bursty packet losses in LSNs for endpoint applications.

### 3.1 System Overview

**Architecture and baseline approach.** Figure 7 plots a high-level view of SatGuard upon today's LSN architecture. SatGuard first adopts a baseline link-local retransmission (denoted as LL-ReTx) approach to fast recover packet losses instead of requiring endpoint retransmission. The baseline LL-ReTx incorporates a *hop-specific* sequence number, called **HSeq**, to facilitate fast loss detection and local recovery. Without loss of generality, we define that packets are forwarded from an upstream node to its downstream node. LL-ReTx requires the upstream node sequentially packs an HSeq in the header of each incoming packet and buffers a copy of this packet in the network interface (*i.e.,* Tx buffer). On the downstream node, LL-ReTx checks the HSeq of arriving packets and sends a link local feedback, denoted as **HAck**, which can instruct the upstream node to detect and retransmit lost packets. Basically, the HAck is sent after each packet arrives, which contains the information that a packet has been successfully received. Once an HAck is received on the upstream node, the corresponding packet can be deleted from the buffered. A packet loss can be detected on the upstream node by examining the HAck mutation (*e.g.,* when receives a larger and discontinuous HAck). When a packet loss is detected,

the upstream node can retransmit the corresponding packet from the buffer locally, instead of sender-side retransmissions.

**Challenges caused by unique LSN characteristics.** While at the first glance, the above baseline approach can attain fast link local retransmission, we highlight three challenges caused by the unique LSN characteristics that can hinder the effectiveness of LL-ReTx.

**(1)** *Buffer invalidation due to LEO dynamics:* Due to the LEO dynamics and endless ground-satellite handovers (*i.e.,* link disconnection and reconstruction), the relationship between the upstream and downstream node is not stable, resulting in LL-ReTx buffer invalidation during handovers. For example, LL-ReTx may fail if the old upstream node with the buffered packets disconnects from the downstream node, and the new upstream node does not has the corresponding packets in its buffer.

**(2)** *Inaccurate loss detection:* The basic LL-ReTx mechanism works well for loss detection and recovery in *one-to-one links*, such as ISLs where one upstream satellite forwards packets to another downstream satellite. However, for *one-to-many shared links* like USLs where the upstream satellite can simultaneously forward packets to a number of downstream user terminals, HAcks from multiple downstream nodes might arrive at the upstream node out of order. In this situation, discontinuous HAck observed by the LL-ReTx upstream node does not necessarily indicate packet losses.

**(3)** *Compatible with endpoint protocols:* Even though lost packets can be recovered by LL-ReTx, the receiver might still receive out-of-order packets, which may mislead the protocol behaviors on endpoints and hinder the user-perceived QoE. For example, some endpoint mechanisms detect packet losses based on the order of received packets. While lost packets have been recovered by LL-ReTx on intermediate nodes, the consequential out-of-order packets observed by the endpoints can still trigger sender-side retransmission, and unexpectedly increase the receiver-side jitter buffer size.

SatGuard addresses the aforementioned challenges by incorporating three key techniques upon the baseline approach, which are marked in green in Figure 7.

**(1)** *Handover-aware buffer migration (§3.2):* To tackle the buffer invalidation caused by LEO dynamics, SatGuard exploits the predictable handover information from the LSN operator and routes buffered content (*e.g.,*the lost packets) from the old upstream node to the new upstream node in time to avoid LL-ReTx failure.

**(2)** *Efficient loss detection via HACK classification (§3.3):* To avoid inaccurate loss detection on one-to-many shared links, Sat-Guard introduces a beam-specific HAck classification method to efficiently detect user-side packet losses in fine-granularity.

**(3)** *In-network order preserving (§3.4):* To mask out-of-order packets caused by LL-ReTx, SatGuard preserves packet orders on ground facilities (*e.g.,*satellite terminals) before forwarding Web traffic to the user equipment (*e.g.,*user laptops).

### 3.2 Handover-aware seamless buffer migration

**Network-layer buffer.** Figure 8 shows a concrete example where the baseline LL-ReTx fails due to the LEO dynamics. Assume packets are forwarded from the satellite to the ground equipment (*e.g.,* GS and dishy). Satellites move from right to left in this example. After a handover, the old upstream satellite (*i.e.,* the left one), which buffers packets, is no longer connected to the ground equipment, resulting in a LL-ReTx failure. SatGuard leverages a network-layer

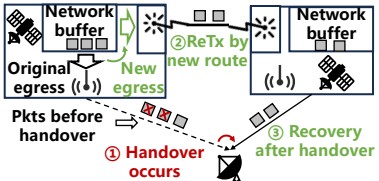

**Figure 8: LL-ReTx failure due to dynamics.**

**Figure 9: Network-layer ReTx can reroute packets in a new path.**

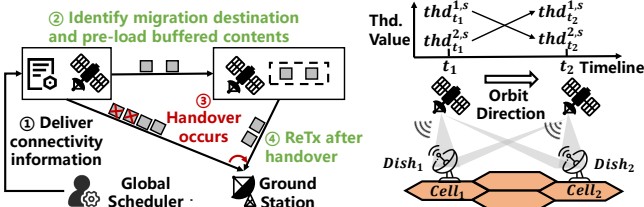

**Figure 10: Ahead-of-handover buffer migration.**

**Figure 11: Beam-based timer setting and update.**

buffer to store packets temporarily. As shown in Figure 9, during the handover, the GS is disconnected from the old satellite (the left one) and then connects to the new satellite (the right one). SatGuard exploits a handover-aware buffer migration strategy which perceives the handover and migrates the buffered packets from the old upstream node to the new upstream node correctly and timely.

**How to find the new satellite?** For SatGuard's buffer migration, one critical problem is how does the old satellite know which new satellite the ground equipment will connect to. To solve this issue, SatGuard exploits a key insight obtained from today's emerging LSN: operators typically leverages a centralized global scheduler to *pre-calculate* and decide ground-satellite connectivity in different time slots [55]. Hence, SatGuard incorporates a connectivity cache on each satellite, which pre-caches the ground-satellite connectivity pre-calculated by the LSN operator. During a handover, the old satellite thus checks its connectivity cache to identify the new satellite that will serve the ground equipment after the handover.

**When and how to migrate buffered contents?** Once the new satellite (*i.e.,* the migration destination) is decided, SatGuard relies on space routing (*e.g.,* [27]) to forward the buffered content to the new satellite. The underlying routing mechanism takes care of the routing calculation, and updates the routing table once the handover happens. Considering that the migration of buffered content can also take a certain amount of time, SatGuard does not start the migration just at the beginning of the handover. Instead, because the time point at which the handover occurs can be predictable according to the connectivity cache, SatGuard starts to forward buffered content ahead of the handover, *i.e.,t* seconds earlier than the handover time. SatGuard sets $t$ approximately to the ratio between buffer size and link capacity. Figure 10 shows an example of such ahead-of-handover migration. First, the LSN operator's global scheduler pre-calculates the connectivity which is pre-cached on satellites in advance. Second, just before the handover occurs, packets that are not acknowledged in the buffer are migrated to the new ingress satellite (*i.e.,* the new upstream node). After the handover, packets can be retransmitted directly from the new satellite.

### 3.3 HAck-classification-based loss detection

For one-to-many shared links (*e.g.,* the radio USL on satellites that serve a number of land-based users), it is difficult to identify packet losses based on discontinuous HAck, since all the downstream nodes may send the HAcks back asynchronously. One approach to locally identify packet losses for LL-ReTx is to set a timer and treat unacknowledged packets after timeout as lost. For example, the LL-ReTx used in standard 802.11 WiFi [10] suggests to set a fixed timer on the WiFi access point to detect packet losses. However, unlike

conventional WiFi which serves users in nearby locations, satellites have a significantly wider communication coverage. The distance between land-based users and satellites varies greatly, resulting in inherent propagation delay differences between users served by a satellite. A fixed timeout value can not effectively identify packet losses in USLs with dynamic time-varying propagation distances. **Beam-based HAck classification.** To effectively detect packet losses on USLs, SatGuard exploits another key insight in LSNs: emerging satellites typically use multiple high-throughput spot-beams to serve geo-distributed users grouped into *cells*, which are terrestrial regions decided by the LSN operator. A cell is attached by at least one spot-beam, and it clusters a number of users in geo-graphically nearby locations. Therefore, SatGuard classifies HAcks from geo-distributed users based on the beam the users are attached to, and set an independent timer for each beam to detect lost packets. In particular, SatGuard inserts a *beam* tag in each HAck to distinguish its group. SatGuard dynamically calculates $thd_t^{k,s}$, the timeout threshold for the cell attached to spot-beam $k$ of satellite $s$ in time $t$, and $thd_t^{k,s} = \frac{2 \times \text{distance}_t^{k_{cell},s}}{c}$, where $\text{distance}_t^{k_{cell},s}$ is the straight-line physical distance from the center of the $k$-attached cell to the satellite $s$ in time $t$, and $c$ is the propagation speed of electromagnetic waves. For each packet sent from $s$ to users in cell $k_{cell}$ in time $t$, if the satellite does not receive HAck for $thd_t^{k,s}$ seconds, the packet is considered lost. Figure 11 plots an example of beam-based timer setting and update. In time $t_1$, the satellite $s$ was closer to $Dish_1$ in $Cell_1$, resulting in $thd_{t_1}^{1,s} < thd_{t_1}^{2,s}$. As the satellite moves, it gets closer to $Dish_2$ in $Cell_2$ and a handover occurs in time $t_2$, thus $thd_{t_2}^{1,s} > thd_{t_2}^{2,s}$.

### 3.4 In-network order preserving

SatGuard adopts a ground-assist order preserving buffer to migrate out-of-order packets caused by LL-ReTx. The buffer only needs to be deployed at the ground facilities (*i.e.,* ground stations or satellite terminals) and does not require modifications on endpoints. **Order identification.** SatGuard adds a flow-level network-layer sequence number (FSeq) to packets before they enter the satellite network (*i.e.,* at the dishy), and does not allow them to be modified by intermediate nodes, so that the correct order can be recognized when they leave lossy satellite environment (*e.g.,* at the GS). **Reordering packets by FSeq.** The basic operation of the buffer is to wait for local retransmitted packets appropriately and forward packets in order. Basically, the reordering buffer checks the FSeq of received packets. Once the FSeq mutation is detected, the packet will be buffered. Otherwise, the packet will be forwarded immediately. The reordering buffer also identifies duplicate packets based on

duplicate FSeq of the flow and discards them. The reordering is executed in flow-level, so there is no head-of-line blocking problem. SatGuard also records the interval of receiving in-order packets and recovery delay of local retransmitted packets to dynamically adjust the maximum buffering time of disordered packets.

## 3.5 Overhead optimization

**Partial traffic processing.** Processing HSeq/HAck and updating the header of packets on satellite routers can involve additional computation overhead. Note that only part of LSN traffic is delay-sensitive. To reduce the overhead on each intermediate node, SatGuard adopts a *partial traffic processing* mechanism, and only delay-sensitive traffic with stringent performance requirements will be processed by SatGuard's local recovery mechanism. In practice, satellite Internet operators can mark the delay-sensitive traffic of high-value customers by configuring the `Traffic Class` option [24] in the standard IP header.

**Dealing with ReTx and HAck loss.** Retransmitted (ReTx) packets may also be lost again. Some previous efforts (*e.g.,* [35]) retransmit a packet multiple times, which involves additional bandwidth overhead. To identify the loss of retransmitted packets, SatGuard updates the HSeq of retransmitted packets. For example, the upstream node sends packets with HSeq from 1 to 4 to the downstream node, and the packet with HSeq=3 is lost. Then the upstream node only receives HAck for 1,2 and 4. In this case, the upstream node retransmits packet with original HSeq = 3, and assigns a new HSeq according to its latest sending order, *i.e.,* HSeq=5 in the retransmitted packet. The unnecessary retransmission caused ny HAck loss can be mitigated by enabling cumulative HAck. Since SatGuard reassigns a new larger HSeq to a retransmitted packet, the HAck in SatGuard is designed to carry the information about the biggest interval of the most recently received consecutive HSeq.

## 3.6 SatGuard Implementation

We have implemented a SatGuard prototype with all the features described above. Since many emerging satellites run Linux-like on-board operating systems (*e.g.,* [3]), we implement the current version of SatGuard upon Linux kernel 5.4. Next we highlight the salient aspects of SatGuard's implementation.

**Exploiting IPv6 extension to embed recovery information in data packets.** To facilitate compatible deployment, we implement SatGuard's network-level HSeq, HAck and FSeq based on the standard IPv6 `Hop-by-Hop` options header [24], which can be processed on each network node. In particular, an HSeq, HAck and FSeq occupies 16 bits in one options header respectively. To implement the in-network functions of SatGuard, we use `lib_netfilterqueue` [12] to capture packets in kernel space for subsequent packet parsing and processing on intermediate nodes.

**Leveraging segment routing to guide fast buffer migration.** We implement SatGuard's buffer migration described in §3.2 based on Segment Routing (SR) [26] which is intrinsically supported by Linux kernel. SR allows a network node to append a header to packets that contain a list of segments, which are instructions that can be executed on subsequent nodes in the network. Therefore, when migrating buffered packets during a handover, SatGuard leverages SR to re-route packets to the predicted next ingress satellite.

# 4 PERFORMANCE EVALUATION

## 4.1 Experiment Setup

We use a recent tool [39] to build a container-based LSN simulator based on real constellation information (*e.g.,* real orbital parameters and distribution of ground stations). The LSN simulator can mimic the LEO dynamics and network behaviors of large-scale LSNs. Each container is a Linux docker instance simulating a network node (*i.e.,* a satellite, ground station or satellite terminal). Moreover, our LSN simulator supports the run of real TCP/IP stack and thus we can load our SatGuard implementation on each simulated network node and load interactive Web traffic in our experiments. We deploy the LSN simulator on two DELL R940xa rack servers and each machine has two Intel Xeon 5217 Processors and 8GB RDIMM.

**Network configurations.** We use our LSN simulator to mimic the network environment of SpaceX's Starlink phase-I constellation which consists of 1584 satellites operating in 72 orbital planes. Further, we configure the packet loss rate in the experimental environment based on our real-world measurements collected from the operational Starlink. As LEO satellites move, we simulate the ground-satellite handovers, and the ground equipment selects a new ingress satellite with the shortest distance [23, 55], which is widely used in today's satellite communication systems. The space segment of the LSN adopts the +Grid connectivity [2, 30], which indicates that each satellite connects to two adjacent satellites in the same orbit (front and rear), and other two satellites in adjacent orbits (left and right). The maximum capacity of the ISL and GSL is configured to 1Gbps. Routing for LSNs is not the focus of this paper, and we use the well-known snapshot routing [27] as the underlying routing mechanism. In snapshot routing, each network node has a pre-calculated valid routing table for each time slot, which indicates how to forward incoming packets on network nodes.

**Approaches of comparison.** We compared the performance of SatGuard with recent state-of-the-art mechanisms for improving network performance in lossy environment: SaTCP [21], Tambur [54], M-PEP [22] and LinkGuard [35]. SaTCP is a link-layer assisted transport enhancement mechanism deployed in the end points in emerging LSNs. SaTCP leverages the handover signal to freeze the congestion window to keep high throughput during handover. The key idea behinds Tambur [54] is to leverage streaming codes to guide receiver to recover lost data in a required guard spaces of packet receptions. M-PEP [22] is a solution based on the well-studied Performance Enhancement Proxy (PEP) technology, which uses in-network proxies to transparently split TCP connections and mitigate the impact of long delay and high loss. We deploy M-PEPs on the ground stations, dishes and their ingress satellites. LinkGuard [35] uses small buffer to achieve link-local retransmission without per-flow state to reduce tail delay in links caused by packet losses. Next we verify the effectiveness of SatGuard from both the network and application perspectives.

## 4.2 Network-level Performance

**Improvement on tail delay.** Figure 12 plots the 99th percentile delay for different communication pairs between populated cities around the world. Since SaTCP only freezes congestion window of the sender during handovers, it can not accelerate the loss recovery and it retransmits lost packets from the sender, resulting high tail

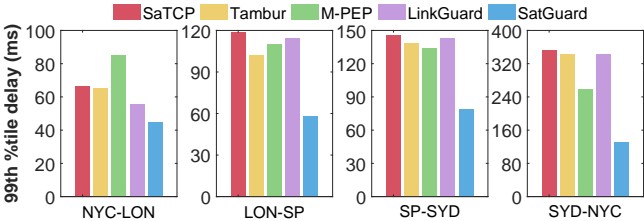

Figure 12: 99th percentile user-perceived end-to-end delay of different communication pairs. NYC: New York City. LON: London. SYD: Sydney. SP: Sao Paulo.

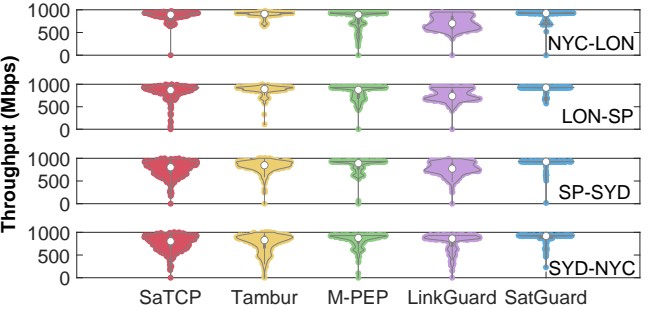

Figure 13: Violin plot of the achievable end-to-end throughput between different communication pairs.

delay up to 351.97 ms for SYD-NYC communication. Although Tambur leverages streaming codes to guide local recovery on the receiver, the required guard spaces of packet receptions in LSNs can not be guaranteed due to consistent packet loss in an error-prone environment, which affects the success rate of local recovery and prolongs the delay up to 342.27 ms. The tail delay in M-PEP ranges from 84.98 ms to 258.10 ms, because M-PEP needs to migrate various state information (*e.g.,* the parameters related to rate control) from the previous ingress satellite to the new ingress satellite before restarting the transmission, which can cost as much as 91.2 ms, depending on the transmission path between the two satellites. After handovers, the buffered packets in the LinkGuard LL-ReTx buffer can not be retransmitted successfully, triggering long-time sender-based recovery and leading to a tail delay up to 342.90 ms. Based on the extended LL-ReTx mechanism, SatGuard leverages network-layer buffer and route updates information to achieve fast buffer migration after handovers, which effectively reduces the tail delay by up to 56.2% for all communication pairs.

**Improvement on end-to-end throughput.** Next, we compare the achievable end-to-end throughput by different approaches. As shown in Figure 13, SaTCP freezes the congestion window during the handover by predicting the moment when the handover occurs. However, packet loss due to interference factors such as bad weather is unpredictable, which makes the throughput in SaTCP less than 500 Mbps in 13.69% of the time. In Tambur, once any packet loss occurs during the guard spaces, lost packets cannot be recovered in the receiver, thus reducing the sending rate to less than 500 Mbps in 12.94% of the time. During the handover, M-PEP needs to resume the transmission after state transition, so the flow frequently goes through the slow start phase. For LinkGuard, although it can effectively recovery packet loss in non-handover scenarios, those lost packets caused by LEO dynamics are still perceived by the sender. Besides, due to unreasonable HAck timeout settings

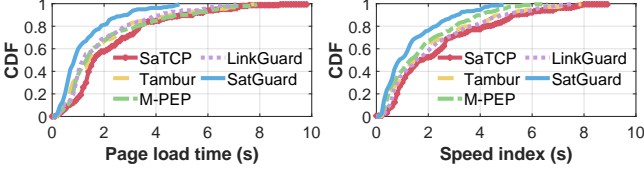

Figure 14: The CDF of page load time and speed index for browsing websites by different loss recovery approaches.

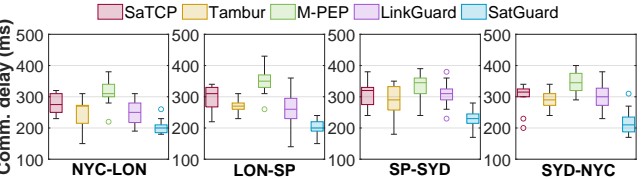

Figure 15: User-perceived one-way WebRTC session delay.

in the one-to-many shared links, the loss detection efficiency is reduced, so the attainable average throughput varies from 719.3 Mbps to 785.2 Mbps. Though integrating handover-aware buffer migration and HAck classification to extend LL-ReTx, SatGuard can locally recover lost packets in various scenarios, maintaining high throughput of more than 800 Mbps for 83.04% of the time.

## 4.3 Application-level Performance

SatGuard **can accelerate Web browsing in LSNs.** We next evaluate the effectiveness of SatGuard on improving the user experience of Web browsing. We use mahimahi [49] to record and replay the workload of browsing all sites in the Alexa Top 200 [5] with different reliability control mechanism. As shown in Figure 14, SatGuard achieves the lowest page load time and speed index. Specifically, the average page load time in SatGuard is 48.3%, 38.1%, 40.2% and 34.4% shorter than that in SaTCP, Tambur, M-PEP and LinkGuard respectively. The achieves speed index in SatGuard is 46.6%/36.2%/25.9% and 38.9% shorter than that in other mechanisms, respectively. The main reason is that SatGuard can recover various losses in time, while (i) SaTCP has to re-request resource from Web servers when any loss occurs, (ii) Tambur can not set appropriate redundancy in such a short flow, (iii) M-PEP has to load resources until state migration completion and transport connection reconstruction, and (iv) LinkGuard behaves similarly to SaTCP in the face of packet losses due to handovers.

SatGuard **can sustain low-delay and smooth WebRTC communications.** Finally, to evaluate the performance improvement of SatGuard on Web-based RTC applications, we use two thinkpad-x1 laptops connect to our LSN simulator as the sender and receiver, and build WebRTC sessions via kurento [11]. We run each RTC session for at least 1 hour and use the WebRTC statistic APIs to capture core QoE metrics including delay, jitter and FPS.

(i) *End-to-end user-perceived communication delay.* Figure 15 plots the communication delay of different sessions. The average communication delay in SatGuard is 213.3 ms, and the delay variation among different city pairs is small, because most lost packets can be recovered locally without passing through long-path retransmission. Therefore, the communication delay in M-PEP also fluctuates very little. However, due to long-term state migration and connection reconstruction during every handover, the communication delay is stable at a high value of about 335.8 ms. Due to the

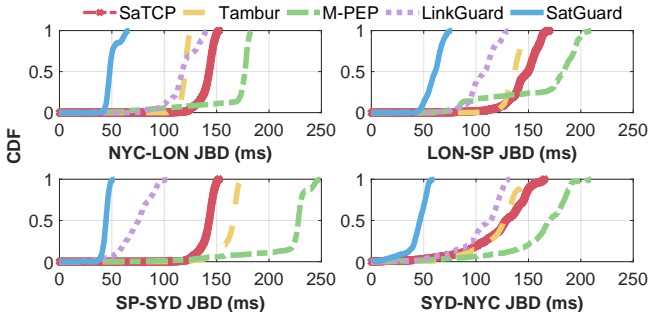

Figure 16: Jitter buffer delay for different WebRTC sessions.

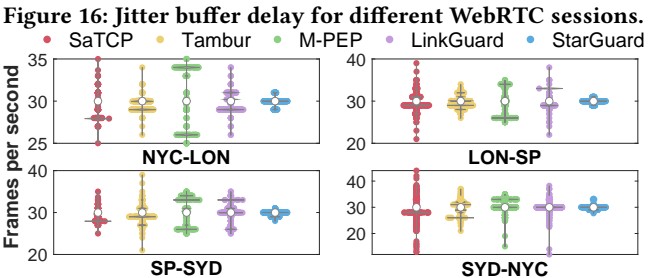

Figure 17: User-perceived FPS in different WebRTC sessions.

lack of local recovery capacity to dynamic packet loss, SaTCP and LinkGuard, in such cases, must endure the impact of sender-based retransmission on the communication delay, which was aggravated with the increase of the communication distance, resulting in an average delay of 298.6 ms and 279.4 ms.

(ii) *Jitter buffer delay (JBD).* As shown in Figure 16, the JBD in M-PEP is more than 150 ms in up to 94.1% of the session period, because frequent long-term state migration causes the receiver to experience large jitter while receiving packets. Tambur and SaTCP cause JBD higher than 100 ms in at least 86.2% and 84.3% of the time because they are difficult to maintain low jitter over a long period of time. Although LinkGuard can achieve low jitter in static scenarios constantly by local retransmission, it fails in dynamic scenarios, causing that the JBD fluctuates greatly, from 52.1 to 130.2 ms. In all communication pairs, the JBD in SATGUARD is less than 73.1 ms because the packet loss caused by handovers can also be effectively recovered.

(iii) *Frames per second (FPS).* As plotted in Figure 17, the FPS of M-PEP is stable at about 33 and 26 most of the time. In fact, we find that the FPS of M-PEP is hopping between the two values frequently, mainly because during the state migration among PEP nodes, a large amount of accumulated frames are played quickly after the transmission is resumed. The FPS in SaTCP is less than 29.5 in 41.6% of time and varies from 12.3 to 43.2. Such fluctuation is mainly because lost frames can not being recovered in time and received frames wait too long in the jitter buffer. Although the average FPS achieved by Tambur and LinkGuard are 29.91 and 29.83 respectively, the standard deviation of Tambur and LinkGundian is 423% and 336% than that of SATGUARD, indicating an unexpected FPS variation which may hinder user experience. The main reason, similar to SaTCP, is that the lost frame information can not be recovered by LinkGuard during handovers and by Tambur when loss occurs in required guard space, and thus discarded in playout. Empowered by handover-aware LL-ReTx, our SATGUARD keeps a stable 30 FPS and guarantees smooth experience for WebRTC.

## 5 RELATED WORK

We briefly discuss efforts related to our work not covered in §2.3.

**Real-world LSN measurement.** With the expansion of the service area, there have been many measurement reports on satellite networks performance. Authors in [48] measure the performance of QUIC [40] and TCP in Starlink and find QUIC can achieve more stable throughput for uploads in such a dynamic environment but has a lower throughput for downloads. Measurement in [36] shows frequent and severe packet losses of up to 50% of packets, which usually occurs during handovers in Starlink. Authors in [55] reveal the existence of Starlink's global scheduler through long-duration measurements, which allocates satellites to terminals every 15 seconds. In [50], authors show the composition of the Starlink access and backbone network through a large number of traceroute results. [29] observes throughput decrease and [56] finds delay increase every 15 seconds respectively. These explorations demystify the network architecture and protocols of emerging LSNs, and help us understand LSN behaviors. Our measurement study in this paper goes one step further, by characterizing the QoE of representative Web services, which are important applications in future LSNs.

**Web application performance and optimization in lossy environment.** Authors in [53] finds that in a wireless environment, inter-RAT (Radio Access Technology) handover caused by user mobility is the most important reason for QoE degradation, which can induce a significant increase in page loading time and speed index, and even frequent Web page loading failures. WiseTrans [60] is a selection mechanism to adaptively switch to the transport protocol that best suits the current network environment based on the detected packet loss rate, round-trip time and bandwidth to reduce delay tail and request completion time, which can be used in collaboration with SATGUARD to reduce the impact of switching transport protocols. Authors in [25] design a WebRTC-compliant video conferencing platform that leverages the larger bandwidth provided by multipath to meet the rising demand for high-resolution video conferencing. SATGUARD can improve transmission performance on each subpath, further improving overall platform performance. Like SaTCP, authors in [43] uses historical handover logs to predict the time of the next handover and quickly restores the sending rate after the handovers. However, this method cannot reduce packet loss recovery delay and still suffers from high communication delay.

## 6 CONCLUSION

This paper proposes SATGUARD, a distributed in-orbit loss recovery mechanism based on link local retransmission. Key ideas behind SATGUARD include (i) exploiting handover information provided by satellite operators to effectively migrating buffers to correct node to prevent buffer invalidation, (ii) leveraging dynamic beam-specific timeout calculation to set the appropriate loss detection parameters for users in different cells to improve recovery efficiency, and (iii) processing out-or-order packets in the ground to make in-network recovery transparent to the upper-layer protocols and applications. The implementation of SATGUARD is based on existing standard protocols, and it is compatible with existing Internet protocol stack. Extensive trace-driven evaluations show that SATGUARD can reduce average page load time for Web browsing and user-perceived communication delay for WebRTC by up to 48.3% and 57.4%.

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
