# OpenReview forum: "SatGuard: Concealing Endless and Bursty Packet Losses in LEO Satellite Networks for Delay-Sensitive Web Applications"
_ACM.org/TheWebConf/2024/Conference — TheWebConf24 Oral_

### Official Review · Reviewer_LxcQ · 2023-11-17

**Novelty:** 5
**Technical Quality:** 6

**Review:**

This paper first measures the network and application performance in Starlink, a production LSN. The measurement  results show that LSN is limited by frequent handovers in LIS. The handovers cause heavy packet loss and latency. To improve the network performance in LSN, this paper propose SatGuard which recover the lost packets by link local retransmission. In LSN simulation environment, SatGuard can improve the LSN performance compared with other LSN mechanisms. I think this paper is interesting and the evaluation is good.

Strengths：

-This paper reveals that handover between satellite is the main factor that influence the network performance in LSN.

-The evaluation shows that SatGuard (by link local retransmission, packet loss detection) can improve the performance of LSN.

Major weaknesses:

-Link local retransmission mechanisms are not new and have been deployed in wireless network. There are not comparisons (mechanism rule or evaluation) between SatGuard and these ones.

**Questions:**

Overall, I like this paper and I thinks its meaningful for the community. And here are some comments which perhaps can be discussed further.

1)	In the evaluation,  the experiments lack prove of the mechanism, such as how the lost packets are recovered,  how the buffer is migrated when handover happens.

2)	There are not description of the experiments environments, such as the packets loss rate, bandwidth, delay.

3)	Link local retransmission mechanisms are not new and have been deployed in wireless network. There are not comparisons (mechanism rule or evaluation) between SatGuard and these ones.

4)	I think some discussion about how the handover affect TCP mechanism (not just loss packet recovery) such as the change of CWND, RTT calculation  can make this paper stronger.

5)	The handover in LSN is similar to some extent with the mobility between cells in cellular network. And there have been some TCP mechanisms([1],[2],[3]) specially designed for the mobility in cellular network. Perhaps the ideas in these mechanisms can further help to improve the data transfer in LSN.

[1]	Leong, Wai Kay, Zixiao Wang, and Ben Leong. "TCP congestion control beyond bandwidth-delay product for mobile cellular networks." In Proceedings of the 13th International Conference on emerging Networking EXperiments and Technologies, pp. 167-179. 2017.

[2]	Lee, Jinsung, Sungyong Lee, Jongyun Lee, Sandesh Dhawaskar Sathyanarayana, Hyoyoung Lim, Jihoon Lee, Xiaoqing Zhu et al. "PERCEIVE: deep learning-based cellular uplink prediction using real-time scheduling patterns." In Proceedings of the 18th International Conference on Mobile Systems, Applications, and Services, pp. 377-390. 2020.

[3]	Abbasloo, Soheil, Yang Xu, and H. Jonathan Chao. "C2TCP: A flexible cellular TCP to meet stringent delay requirements." IEEE Journal on Selected Areas in Communications 37, no. 4 (2019): 918-932.

**Reviewer Confidence:**

3: The reviewer is confident but not certain that the evaluation is correct

**Scope:**

4: The work is relevant to the Web and to the track, and is of broad interest to the community

---

### Official Review · Reviewer_9EwC · 2023-11-20

**Novelty:** 4
**Technical Quality:** 3

**Review:**

This paper proposes an in-network link layer retransmission scheme for LEO satellite network such as Starlink. The approach is called SatGuard.

I noticed a few problems in the motivation and the claimed properties of Starlink:

1. "Upper layer applications are marching to the new era of Web-3.0."
    * No, they are really not. These are hyperbolic marketing claims of web-3.0 proponents. It's not a good idea to repeat them in a scientific publication.
   * Even if they were, it would really not be relevant to the topic of this paper.
   * A link layer retransmission as proposed here does not really have a strong connection with web topics. The mechanisms would apply to *any* type of communication – maybe the paper would be a better fit for a satellite communication publication venue?
2. Starlink's "endless and bust packet losses".
   * You seem to exaggerate the performance problems, specifically the packet loss rates. Other measurements and publications report a less problematic network performance, e.g.:
      - https://www.netforecast.com/wp-content/uploads/FixedWireless_LEO_CableComparisonReport_NFR5148-1.pdf
      - https://blog.apnic.net/2022/11/28/fact-checking-starlinks-performance-figures/
   * From my own experience, packet loss occurs at low rates. Higher rates are often caused by sub-optimal antenna position, FOV blocking, or antenna mobility.

With respect to the proposed technical approach, I have a few comments:

3. You did not really analyze how current LEO link layers work. This would have been a good basis for your design.
4. Considering the current low (or at least acceptable) loss rates, inventing a handover-aware buffer migration scheme is obviously quite complex and costly. You do not discuss this a lot.
5. The assumptions for the partial traffic processing feature (that you could QoS classification) is unrealistic in today's Internet. Diffserv is not used inter-domain, and given ubiquitous encryption, it is hard/impossible to classify traffic in the network.
6. It's good that you performed application-level performance tests. The discussion could be more technical. E.g., what protocols were actually used (RTP, I assume?). How did the packet loss without your scheme affect the FPS?

**Questions:**

1. Can you reconsider your claims on the unreliability of current the Starlink network? There may be other LEO networks that match those claims.
2. Can you base your design on actual design of current LEO link layer protocols?
3. Can you discuss the complexity issue?
4. Can you discuss the impact on WebRTC communication with more technical depth?

**Ethics Review Description:**

no issues

**Reviewer Confidence:**

2: The reviewer is willing to defend the evaluation, but it is likely that the reviewer did not understand parts of the paper

**Scope:**

2: The connection to the Web is incidental, e.g., use of Web data or API

---

### Official Review · Reviewer_dZp2 · 2023-11-22

**Novelty:** 4
**Technical Quality:** 5

**Review:**

This paper proposes Satguard which is an approach to achieve fast packet loss recovery for low earth orbit satellite constellations. The paper performs a measurement study on Starlink to show persistent and periodic packets losses which correlate with satellite handovers. Satguard achieves fast loss recovery through three mechanisms (i) advance sender buffer migration to future ingress satellite before a handover, (ii) early loss detection through beam specific timers and  (iii) in-network ordered packet delivery. Evaluations are conducted on a simulated satellite network constellation and shows that Satguard minimizes end-to-end delay, improves web page load times and sustains stable high frame rate in WebRTC streams.

### Pros:
- Valuable insights from measurement study.
- Paper agrees to open-source Satguard.
- Paper is on a timely topic and is well written for most parts.

### Cons:
- The overheads introduced by Satguard are not evaluated.
- Some aspects of the design such as connectivity cache are glossed over.

**Questions:**

- The in-network packet ordering mechanism on the ground terminal functions like a jitter buffer but for many concurrent flows. What is the additional latency introduced by it to order the packets before delivery to sender? Furthermore, what is the overhead and impact of per flow state it needs to maintain?

- How is the connectivity cache kept updated? Even if a global scheduler is deployed, are there guarantees that pre-calculated connectivity plans will not change dynamically at runtime?

- While the paper notes the additional overhead of marking packets, it doesn’t receive much attention in the evaluation section so it remains unclear whether the approach is practical. Furthermore, how can satellite ISPs reliably determine traffic class to annotate packet headers?

- It seems Satguard assumes that packets can not be lost between intermediate nodes? It is unclear if this is a reasonable assumption to make.

### Writing nits.
- **Sec 1:** statue quo -> status quo
- **Sec 1:** *due to the high LEO dynamics* . “High LEO dynamics” is vague, while the context helps to explain what is meant, but it is better to be clear what this refers to in the intro.
- **Sec 2.2**: *523.3% of medium value* -> 523.3% of median value
- **Sec 4.2**: effectively recovery packet loss -> effectively recover packet loss

**Reviewer Confidence:**

2: The reviewer is willing to defend the evaluation, but it is likely that the reviewer did not understand parts of the paper

**Scope:**

4: The work is relevant to the Web and to the track, and is of broad interest to the community

---

### Official Review · Reviewer_PyHy · 2023-11-30

**Novelty:** 5
**Technical Quality:** 6

**Review:**

In this paper, the authors address the challenge of delivering delay-sensitive Web applications via low earth orbit satellites. The authors propose several techniques to comply with LEO satellites characteristics and to reduce delay. The analysis is supported a prototype and trace-driven emulation.

I like the perspective of the authors that delay, even though in low orbit, is less of an issue compared to unstable links.

Without doubt, the paper is nicely written und includes clear thoughts. What would be nice to see is a much better articulation about the novelty of the proposal.

Some comments:

(1) The in-network caching approach is related to types of prior work. First, anything that relates to information-centric (or named-data) networking. Second, approaches that try to cope with IP mobility. In particular, Fast Mobile IPv6 had a very similar idea, i.e., sending buffered content ahead to bridge handover gaps. How would you compare from a principle design perspective?

(2) The authors propose to use some kind of p2p communication between the satellites (i.e., spacerouting). I'm wondering how realistic is this to implement.

(3) The proposal seems to require synchronized clocks to align with the GS schedule. What about clock drift?

(4) The general problem of lossy links is also prevalent in IoT networks. What could we learn from proposal in this domain?

Editorial remarks:

* Figure 17: StarGuard should probably be SatGuard.

**Questions:**

See above.

**Reviewer Confidence:**

3: The reviewer is confident but not certain that the evaluation is correct

**Scope:**

3: The work is somewhat relevant to the Web and to the track, and is of narrow interest to a sub-community

---

### Decision · Program_Chairs · 2024-01-22

**Decision:**

Accept (Oral)

**Comment:**

The paper proposes SatGuard, addressing packet loss challenges in low earth orbit (LEO) satellite constellations like Starlink. SatGuard employs three mechanisms for fast loss recovery, supported by a measurement study on Starlink, simulations, and an open-source implementation. Reviewers appreciate the paper's insights, well-written nature, and timely relevance to a significant problem. However, some reviewers raised concerns around the proposal's novelty and its realistic implementation. Some reviewers question the exaggerated portrayal of Starlink's performance issues and raise doubts about the proposed link layer retransmission scheme's practicality.

 The work unboubtedly provides valuable insights from a measurement study on Starlink, but it may overlap with more recent studies, such as:
 Aravindh Raman, Matteo Varvello, Hyunseok Chang, Nishanth Sastry, and Yasir Zaki. 2023. Dissecting the Performance of Satellite Network Operators. Proc. ACM Netw. 1, CoNEXT3, Article 15 (December 2023), 25 pages. https://doi.org/10.1145/3629137

 Nevertheless, SatGuard's contribution to fast packet loss recovery is deemed relevant and well-timed by the reviewers. The paper is well-written, and the application-level performance tests provide technical insights. The use of simulations to evaluate SatGuard's impact on end-to-end delay and web page load times is acknowledged.

 The authors have engaged with the reviewers during the rebuttal phase, and presented ways in which they plan to address the feedback they received, and thus improve their submission.
 I would recommend conditionally accepting this paper -- I would probably encourage appointing a shepherd to oversee the camera ready, especially since (from the rebuttal phase) I understand that some reviewers might want to actually see new experiments being performed. Specifically, I'd like that the authors clarify and articulate the novelty of the proposal more explicitly, and that they address concerns about the practicality and complexity of the proposed handover-aware buffer migration scheme.